# Metabolic Profile Variations along the Differentiation of Human-Induced Pluripotent Stem Cells to Dopaminergic Neurons

**DOI:** 10.3390/biomedicines10092069

**Published:** 2022-08-24

**Authors:** Emma Veronica Carsana, Matteo Audano, Silvia Breviario, Silvia Pedretti, Massimo Aureli, Giulia Lunghi, Nico Mitro

**Affiliations:** 1Department of Medical Biotechnology and Translational Medicine, University of Milan, 20054 Milan, Italy; 2Department of Pharmacological and Biomolecular Sciences, University of Milan, 20122 Milan, Italy

**Keywords:** iPSCs, dopaminergic neurons, mass spectrometry, metabolism, neuronal differentiation

## Abstract

In recent years, the availability of induced pluripotent stem cell-based neuronal models has opened new perspectives on the study and therapy of neurological diseases such as Parkinson’s disease. In particular, P. Zhang set up a protocol to efficiently generate dopaminergic neurons from induced pluripotent stem cells. Although the differentiation process of these cells has been widely investigated, there is scant information related to the variation in metabolic features during the differentiation process of pluripotent stem cells to mature dopaminergic neurons. For this reason, we analysed the metabolic profile of induced pluripotent stem cells, neuronal precursors and mature neurons by liquid chromatography–tandem mass spectrometry. We found that induced pluripotent stem cells primarily rely on fatty acid beta-oxidation as a fuel source. Upon progression to neuronal progenitors, it was observed that cells began to shut down fatty acid β-oxidation and preferentially catabolised glucose, which is the principal source of energy in fully differentiated neurons. Interestingly, in neuronal precursors, we observed an increase in amino acids that are likely the result of increased uptake or synthesis, while in mature dopaminergic neurons, we also observed an augmented content of those amino acids needed for dopamine synthesis. In summary, our study highlights a metabolic rewiring occurring during the differentiation stages of dopaminergic neurons.

## 1. Introduction

In the field of neuroscience, one of the main limits resides in the lack of suitable experimental models. Before the birth of induced pluripotent stem cells (iPSCs) technology [1,2], animal models, primary neural cells and immortal cell lines have contributed to our understanding of the nervous system both in physiological and pathological conditions. In particular, great efforts have been made to develop models that better recapitulate diseases of the central nervous system. Among these pathologies, one of the most common is Parkinson’s disease (PD), a neurodegenerative disorder characterised by the loss of dopaminergic neurons within the substantia nigra pars compacta [3,4,5]. To date, the causes that lead to neuronal death are largely unknown, and no effective cure exists [6]. Many cellular and animal models have been generated to study PD, but they present several limitations [7,8,9,10]. Concerning the animal models, interspecies differences are obstacles to the complete recapitulation of disease phenotypes, thus causing a high failure rate in the development of drugs. The use of primary cells is hampered by the impossibility of isolating viable neurons from adult brains.. On the other hand, even if cell lines generated from brain tumours can be easily cultured, their use is limited by their oncogenic characteristics. Studies on the postmortem human brain are limited by its rapid deterioration since neural cells are extremely sensitive to oxygen and blood supply. In recent years, the availability of iPSC-based models has opened new perspectives on the study of neurological diseases [11,12]. iPSCs are generated by reprogramming human somatic cells, thus avoiding the species differences associated with the use of animal models. In addition, they are pluripotent, similar to embryonic stem cells (ESCs), meaning they are able to differentiate into all cell types of the human body under appropriate culture conditions [1,13,14]. Moreover, iPSCs reprogrammed from patient somatic cells retain their original genomic features, such as gene mutations and chromosome abnormalities, and can, therefore, be used to study the effects of genomic defects on cell functions [15,16,17,18]. Nowadays, the availability of specific differentiation protocols allows to generate definite subpopulations of neurons, including dopaminergic (DA) ones, providing a significant step forward in the study of PD. In particular, P. Zhang and colleagues [19] have optimised an efficient and reproducible protocol for the generation of dopaminergic neurons (DA neurons) from human iPSCs [20]. The protocol consists of the generation of floor plate (FP) precursor cells by the activation of the canonical Wnt and sonic hedgehog signalling. After 11 days of differentiation, human iPSCs are efficiently converted to FP precursors. Around day 25 of differentiation, the FP precursor cells are specified to DA neurons, while after 60 days of differentiation, mature DA neurons are obtained [19]. The differentiation process of these cells has been widely investigated [21]; nevertheless, there is scant information related to the variation in metabolic features during the differentiation process of pluripotent stem cells to mature neurons. To address this aim, we performed a metabolomics analysis of cells at different stages of differentiation. We found that iPSCs are metabolically different from neurons at all the analysed time points. In particular, pluripotent stem cells mainly rely on fatty acid oxidation as energetic fuel, contrary to what happens during differentiation, in which the oxidation of glucose becomes the main energy source. Moreover, during differentiation, we found a marked increase in the content of amino acids involved in the production of tricarboxylic acid cycle intermediates. Interestingly, in mature neurons, we also assisted in an increase in amino acids needed for dopamine synthesis. Our results represent a useful background for the interpretation of metabolic alterations occurring in neurodegenerative disorders.

## 2. Materials and Methods

### 2.1. Materials

Commercial chemicals were of the highest purity available; common solvents were distilled before use, and water was doubly distilled in a glass apparatus. Calcium magnesium free (CMF)-PBS, bovine serum albumin, donkey serum, rabbit polyclonal anti-GAPDH antibody (RRID: AB_796208), ROCK inhibitor, SB431542, SAG, purmorphamine, CHIR99021, cAMP and L-ascorbic acid were from Sigma-Aldrich (St. Louis, MO, USA). L-glutamine, penicillin/streptomycin (10,000 units/mL) and D-MEM were from EuroClone (Paignton, UK). Mouse anti-neurofilament H (RRID: AB_10694081), rabbit anti-MAP2 (RRID: AB_10693782), mouse anti-tau (RRID: AB_10695394), mouse anti-β3-tubulin (RRID: AB_1904176), mouse anti-nestin (RRID: AB_2799037), rabbit anti-Sox2 (RRID: AB_1658242), rabbit anti-Nanog (RRID: AB_1658242), mouse anti-TRA-1-81 (RRID: AB_1658242), mouse anti-TRA-1-60 (RRID: AB_1658242) and goat anti-rabbit HRP-conjugated antibodies (RRID: AB_2099233) were from Cell Signalling Technology (Danvers, MA, USA). Mouse anti-TH (RRID: AB_628422) antibody was from Santa Cruz Biotechnology (Dallas, TX, USA). The chemiluminescence kit for immunoblotting was from Cyanagen (Bologna, Italy). Neurobasal medium 1×, B27 supplement, N2 supplement, KSR supplement, Geltrex, Essential 8 Medium, Accutase, 100× nonessential amino acid, 100× 2-mercaptoethanol, 100× Glutamax, goat anti-mouse HRP-conjugated (RRID: AB_228307), anti-rabbit AlexaFluor568 (RRID: AB_2534017), anti-mouse AlexaFluor568 (RRID: AB_2534013) and anti-rabbit AlexaFluor488 (RRID: AB_2576217) were from ThermoFisher Scientific (Waltham, MA, USA). The 4–20% Mini-PROTEAN^®^ TGX™ Precast Protein Gels, turbo polyvinylidene difluoride (PVDF) Mini-Midi membrane and DC™ protein assay kit were from BioRad (Hercules, CA, USA). LDN-193189 was from Reprocell (Beltsville, MD, USA). FGF-8b, BDNF, GDNF and TGF-3β were from Peprotech (London, UK).

### 2.2. iPSC Culture

Human iPSC clonal line obtained from fibroblasts of a healthy subject was purchased from Coriell Institute (AICS-0022-037). Parental hiPSC line (WTC/AICS-0 at passage 33) derived from fibroblasts was reprogrammed using episomal vectors (OCT3/4, shp53, SOX2, KLF4, LMYC and LIN28). iPSCs were grown in Geltrex-coated (1% for 1 h at 37 °C) six-well plates and cultured in complete Essential 8 Medium. At 80–90% confluence, cells were passaged using Accutase (3 min 37 °C) and plated at a density of 104 cells/cm^2^ in complete Essential 8 Medium supplemented with 10 µM ROCK inhibitor for 24 h.

### 2.3. Differentiation of iPSC into Dopaminergic Neurons

iPSCs were differentiated into DA neurons according to the protocol described by Zhang et al. [19]. Cells were cultured in proper media supplemented with specific factors at proper concentrations as follows. Day 0: KSR differentiation medium (81% DMEM, 15% KSR, 100× 1% nonessential amino acids, 100× 1% 2-mercaptoethanol, 100 U/mL penicillin and 100 µg/mL streptomycin) supplemented with 10 μM SB431542 and 100 µM LDN-193189. Days 1 and 2: KSR differentiation medium supplemented with 10 μM SB431542, 100 nM LDN-193189, 0.25 μM SAG, 2 μM purmorphamine and 50 ng/mL FGF8b. Days 3 and 4: KSR differentiation medium supplemented with 10 μM SB431542, 100 nM LDN-193189, 0.25 μM SAG, 2 μM purmorphamine, 50 ng/mL FGF8b and 3 μM CHIR99021. Days 5 and 6: 75% KSR differentiation medium and 25% N2 differentiation medium (97% DMEM, 100× 1% N2 supplement, 100 U/mL penicillin and 100 µg/mL streptomycin) supplemented with 100 nM LDN-193189, 0.25 μM SAG, 2 μM purmorphamine, 50 ng/mL FGF8b and 3 μM CHIR99021. Days 7 and 8: 50% KSR differentiation medium and 50% N2 differentiation medium supplemented with 100 nM LDN-193189 and 3 μM CHIR99021. Days 9 and 10: 25% KSR differentiation medium and 75% N2 differentiation medium supplemented with 100 nM LDN-193189 and 3 μM CHIR99021. Days 11 and 12: B27 differentiation medium (95% neurobasal medium, 50× 2% B27 supplement, 1% Glutamax, 100× 100 U/mL penicillin and 100 µg/mL streptomycin) supplemented with 3 μM CHIR99021, 10 ng/mL BDNF, 10 ng/mL GDNF, 1 ng/mL TGF-b3, 0.2 mM ascorbic acid and 0.1 mM cyclic AMP. From day 13 to the end of differentiation: B27 differentiation medium supplemented with 10 ng/mL BDNF, 10 ng/mL GDNF, 1 ng/mL TGF-β3, 0.2 mM ascorbic acid and 0.1 mM cyclic AMP. After 20 days of differentiation, cells were split using Accutase (3 min 37 °C) on Geltrex-coated plates at a density of 2 × 10^5^ cells/cm^2^. The medium was changed every day.

### 2.4. Protein Determination

Protein concentration of samples was assessed with the DC™ protein assay kit according to manufacturer’s instructions, using bovine serum albumin (BSA) at different concentrations as standard.

#### 2.4.1. Immunoblotting

Immunoblotting on total cell lysates from iPSCs and DA neurons at days 24, 45 and 60 of differentiation were performed using standard protocols. Aliquots of proteins were mixed with Laemmli buffer (0.15 M DTT, 94 mM Tris-HCl pH 6.8, 15% glycerol, 3% *w*/*v* SDS, 0.015% blue bromophenol) and heated for 5 min at 95 °C. Proteins were separated on 4–20% polyacrylamide gradient gels and transferred to polyvinylidene difluoride (PVDF) membranes by electroblotting. PVDF membranes were incubated in blocking solution (5% nonfat dry milk (*w*/*v*) in TBS-0.1% tween-20 (*v*/*v*)) at 23 °C for 1 h under gentle shaking. Subsequently, PVDF membranes were incubated overnight at 4 °C with primary antibodies diluted in blocking solution. The day after, PVDF membranes were incubated for 1 h at 23 °C with secondary HRP-conjugated antibodies diluted in blocking solution. PVDF membranes were scanned using the chemiluminescence system Alliance Mini HD9 (Uvitec, Cambridge, UK), and band intensity was quantified using ImageJ software (v2.1.0/1.53c). The following primary antibodies were used for immunoblotting: monoclonal mouse anti-TH (dilution: 1:2500), polyclonal rabbit anti-MAP2 (dilution: 1:1000), monoclonal mouse anti-tau (dilution: 1:1000), monoclonal mouse anti-neurofilament H (dilution: 1:1000), monoclonal mouse anti-β3-tubulin (dilution: 1:1000), monoclonal mouse anti-nestin (dilution: 1:1000) and polyclonal rabbit anti-GAPDH (dilution: 1:10,000). The following secondary antibodies were used: goat anti-rabbit HRP-conjugated (1:2000) and goat anti-mouse HRP-conjugated (dilution: 1:2000).

#### 2.4.2. Immunofluorescence Staining

iPSCs and DA neurons at days 24, 45 and 60 of differentiation were plated at a density of 2 × 10^4^ and 10^5^ cells/cm^2^, respectively, on 24 mm cover glasses precoated with Geltrex (1%, 1 h 37 °C). Cells were fixed in 4% paraformaldehyde in PBS for 20 min at 23 °C. Cells were then permeabilised in 0.2% Triton X-100 in PBS for 10 min at 23 °C and blocked in 1% BSA/2% donkey serum in PBS for 1 h at 23 °C. Cells were incubated for 2 h at 23 °C with primary antibodies diluted in 0.25% BSA/0.5% donkey serum in PBS and 1 h at 23 °C with secondary antibodies conjugated to Alexa Fluor, diluted in the same solution. Nuclei were stained by incubation with Hoechst solution (2 μg/mL in PBS) for 5 min at 23 °C. The following antibodies were used: monoclonal mouse anti-β3-tubulin (dilution: 1:200), monoclonal mouse anti-TH (F-11) (dilution: 1:50), polyclonal rabbit anti-MAP2 (dilution: 1:50), monoclonal mouse anti-nestin (dilution: 1:250), monoclonal mouse anti-neurofilament H (dilution: 1:200), monoclonal rabbit anti-Sox2 (dilution 1:500), monoclonal rabbit anti-Nanog (dilution 1:500), monoclonal mouse anti-TRA-1-81 (dilution 1:500), monoclonal mouse anti-TRA-1-60 (dilution 1:500), polyclonal anti-mouse AlexaFluor568, polyclonal anti-rabbit AlexaFluor568 and polyclonal anti-rabbit AlexaFluor488. Coverslips were mounted with Dako Fluorescent Mounting Medium (Agilent Technologies, Santa Cruz, CA, USA). Images were taken using an Olympus BX50 Upright Fluorescence Microscope fluorescence with a fast high-resolution camera (Colorview 12).

#### 2.4.3. Targeted Metabolomics Analysis

Metabolomic data were obtained by liquid chromatography coupled to tandem mass spectrometry. We used an API-3500 triple quadrupole mass spectrometer (AB Sciex, Framingham, MA, USA) coupled with an ExionLC™️ AC System (AB Sciex, Framingham, MA, USA). iPSCs and DA neurons at days 24, 45 and 60 of differentiation were extracted using a tissue lyser for 1 min at maximum speed in 250 µL of ice-cold methanol: water: acetonitrile 55:25:20 containing [U-13C6]-glucose 1 ng/µL and [U-13C5]-glutamine 1ng/µL as internal standards (Merk Life Science, Milan, Italy). Lysates corresponding to 1.5 × 10^6^ cells were spun at 15,000 g for 15 min at 4 °C. Samples were then dried under N2 flow at 40 °C and resuspended in 125 µL of ice-cold methanol/water/acetonitrile 55:25:20 for subsequent analyses.

Amino acids, their derivatives and biogenic amine quantification were performed through previous derivatisation. Briefly, 25 µL out of 125 µL of samples were collected and dried separately under N2 flow at 40 °C. Dried samples were resuspended in 50 µL of phenyl-isothiocyanate (PITC, Merk Life Science, Milan, Italy), EtOH, pyridine and water 5%:31.5%:31.5%:31.5% and then incubated for 20 min at RT, dried under N2 flow at 40 °C for 90 min and finally resuspended in 100 µL of 5mM ammonium acetate in MeOH: H_2_O 50:50. Quantification of different amino acids was performed by using a C18 column (Biocrates, Innsbruck, Austria) maintained at 50 °C. The mobile phases for positive ion mode analysis were phase A: 0.2% formic acid in water and phase B: 0.2% formic acid in acetonitrile. The gradient was T0: 100% A, T5.5: 5% A and T7: 100% A with a flow rate of 500 µL/min. All metabolites analysed in the described protocols were previously validated by pure standards, and internal standards were used to check instrument sensitivity.

Quantification of energy metabolites and cofactors was performed by using a cyanophase LUNA column (50 mm × 4.6 mm, 5 µm; Phenomenex, Bologna, Italy) by a 5.5 min run in negative ion mode with two separated runs. Protocol A: mobile phase A was water, phase B was 2 mM ammonium acetate in MeOH, and the gradient was 10% A and 90% B for the analysis, with a flow rate of 500 µL/min. Protocol B: mobile phase A was water, phase B was 2 mM ammonium acetate in MeOH, and the gradient was 50% A and 50% B for the analysis, with a flow rate of 500 µL/min.

Acylcarnitines quantification was performed on the same samples by using a Varian Pursuit XRs Ultra 2.8 Diphenyl column (Agilent, Milan, Italy). Samples were analysed by a 9 min run in positive ion mode. Mobile phases were A: 0.1% formic acid in H20; B: 0.1% formic acid in MeOH; and the gradient was T0: 35% A, T2.0: 35% A, T5.0: 5% A, T5.5: 5% A, T5.51: 35% A and T9.0: 35% A, with a flow rate of 300 µL/min.

MultiQuant™ software (version 3.0.3, AB Sciex, Framingham, MA, USA) was used for data analysis and peak review of chromatograms. Raw areas were normalised by the areas’ median. Obtained data were then compared to relative controls and expressed as fold change. Obtained values were considered as scaled metabolite levels. Data processing and analysis were performed by MetaboAnalyst 5.0 web tool [22].

#### 2.4.4. Statistical Analysis

Statistical analysis for metabolomics data was performed by taking advantage of MetaboAnalyst 5.0 webtool [22]. For multiple comparisons of experimental groups with a normal distribution, a one-way analysis of variance (ANOVA) accompanied by Tukey’s HSD (honestly significant difference) post hoc test was used.

## 3. Results

### 3.1. Characterisation of iPSCs and iPSC-Derived Dopaminergic Neurons

iPSCs pluripotency was assessed by immunofluorescence analysis against pluripotency markers such as Sox2 and Nanog, key transcriptional regulators highly expressed in pluripotent cells and TRA-1-81 and TRA-1-60, different proteoglycan epitopes on variants of the same protein, podocalyxin. As shown in Figure 1, iPSCs expressed all these pluripotency markers.

Subsequently, iPSCs were differentiated into dopaminergic neurons according to P. Zhang’s protocol and analysed at different stages of differentiation in order to characterise the metabolism associated with the different cellular populations. In particular, cells were harvested at 24, 45 and 60 days of differentiation.

As shown by immunoblotting and immunofluorescence analyses (Figure 2 and Figure 3), at day 24 of differentiation, the cellular population was mainly represented by neuronal precursors, as they were characterised by a high expression of nestin (Figure 3), a neural precursor marker, and by a low amount of the main neuronal markers neurofilament H (NFH), βIII-tubulin (TUJ1) and TAU (Figure 2 and Figure 3). Concerning microtubule-associated protein 2 (MAP2), at this stage, cells expressed mainly the low-weight isoform MAP2C, typically present during the early stages of neuronal differentiation in vitro [23,24]. Moreover, these neuronal precursors did not exhibit the features of DA neurons, as they did not express tyrosine hydroxylase (TH), the enzyme required for dopamine synthesis (Figure 2 and Figure 3).

At day 45 of differentiation, the neurons appeared to be more mature, as demonstrated by the increased expression of NFH, TUJ1 and TAU and by the presence of the high-weight MAP2 isoforms MAP2A and B (Figure 2). Moreover, these neurons were characterised by the reduced expression of nestin and started to express the dopaminergic marker TH (Figure 2 and Figure 4).

Finally, at day 60 of differentiation, a fully differentiated population of neurons was obtained, with a high proportion of DA neurons (Figure 2 and Figure 5).

### 3.2. Metabolic Profile at Different Stages of Neuronal Differentiation

To examine how metabolism changes during neuronal in vitro differentiation, we carried out liquid chromatography–tandem mass spectrometry (LC-MS/MS)-based profiling on iPSCs and DA neurons at 24, 45 and 60 days of differentiation. As shown by principal component analysis in Figure 6, the metabolic profile significantly changed during the differentiation process and, in particular, among iPSCs, neuronal precursors (24 days of differentiation) and mature neurons (60 days of differentiation). Notably, between neurons at day 45 and day 60 of differentiation, no significant differences in the metabolic profile were observed. In addition, the heatmap of the metabolites revealed striking differences relative to glycolysis, the tricarboxylic acid (TCA) cycle, amino acids and fatty acid oxidation (Figure 7).

#### 3.2.1. Glycolysis, Pentose–Phosphate Pathway and Tricarboxylic Acid Cycle

Concerning glycolysis (Figure 8), a significant reduction in glucose-6-phosphate (glucose-6-P) was observed during differentiation and, in particular, in neuronal precursors at day 24 of differentiation with respect to iPSCs, while no significant changes were detected in glucose levels. Interestingly, the levels of the glycolysis intermediates dihydroxyacetone phosphate/glyceraldehyde 3-phosphate (DHAP/GAP) and phosphoenolpyruvate (PEP), as well as lactate, significantly increased during differentiation until day 45, then held steady between days 45 and 60. We also observed that the levels of ribose-5-phosphate/xylulose-5-phosphate/ribulose-5-phosphate (R-X-Ru-5P), generated by the pentose–phosphate pathway, progressively increased during the neuronal differentiation process.

During neuronal differentiation, we also observed an increase in several intermediates of the TCA cycle such as acetyl coenzyme A (acetyl-CoA), α-ketoglutarate, fumarate, succinate and malate (Figure 8). In particular, acetyl-CoA significantly increased at the last day of differentiation with respect to iPSCs, while α-ketoglutarate, fumarate, succinate and malate were significantly higher in all the differentiation time points with respect to pluripotent stem cells. Only citrate showed a different trend, increasing at day 24 of differentiation compared with iPSCs, with a subsequent reduction at days 45 and 60 of differentiation. These latter results are in line with the notion that citrate may exit the mitochondria to sustain cholesterol and fatty acid biosynthesis. Together, these results indicate that during differentiation from iPSCs towards DA neurons, glucose sustains both lactate production and flux in the TCA cycle for anabolic functions (i.e., fatty acids and cholesterol biosynthesis).

#### 3.2.2. Fatty Acid β-Oxidation

The measurement of different acylcarnitines is an indirect readout to assess the fatty acid β-oxidation pathway (Figure 9). The levels of free carnitine (C0) resulted in very low iPSCs, which gradually increased until day 45 of differentiation, keeping constant during neuronal maturation. A similar trend had emerged for octadecenoylcarnitine (C18:1-carnitine) and hexadecenoylcarnitine (C16:1-carnitine), with a significant increase at days 45 and 60 of differentiation with respect to iPSCs and neuronal precursors. On the other hand, ocatadecanoyl-(C18-), hexadecanoyl-(C16-) and acetyl-(C2-) carnitines were highly represented in pluripotent stem cells and drastically decreased during differentiation, starting from neuronal precursors. Moreover, the C2/C0 ratio, an indicator of fatty acid β-oxidation, and the activity of the fatty acid shuttle into the mitochondria carnitine palmitoyl transferase 1 (Cpt-1), assessed by the ratio (C16 + C18)/C0, were higher in iPSCs and decreased during the differentiation process. Finally, propionyl (C3-) and butyryl (C4)-carnitine levels fluctuated during neuronal differentiation, showing an increase at day 24 with respect to iPSCs and again a reduction during neuronal maturation at days 45 and 60. Altogether, these results indicate that iPSCs have increased fatty acid oxidation that declined progressively during the neuronal differentiation process.

#### 3.2.3. Amino Acids

Another class of metabolites that changed considerably during the differentiation from iPSCs to mature DA neurons is the amino acids. All identified amino acids are reported in Figure 10. In particular, alanine, glutamine and methionine drastically decreased from day 24 of differentiation and remained low in mature neurons at days 45 and 60, while lysine underwent a greater decrease in mature neurons. Otherwise, asparagine, tryptophan, tyrosine, valine, proline, isoleucine and threonine levels were low in iPSCs and significantly increased during differentiation, starting from the stage of neuronal precursors, differently from phenylalanine and threonine that significantly increased in mature neurons and not in precursors. A less linear trend was observed for asparagine, arginine and glutamic acid: they increased in neuronal precursors at day 24 with respect to iPSCs and decreased in mature neurons at days 45 and 60 of differentiation. Glycine levels were high in pluripotent stem cells and significantly decreased in neuronal precursors and DA neurons at day 45 of differentiation, finally increasing again in mature neurons at day 60. No significant differences were observed concerning serine and histidine levels. These data clearly indicate that some amino acids such as alanine, glutamine and methionine, which are intermediates of metabolic pathways, are consumed during differentiation, while others such as phenylalanine and tyrosine, which are involved in dopamine biosynthesis, significantly increased only in mature dopaminergic neurons.

## 4. Discussion

Over the last decade, changes in metabolism during differentiation from iPSCs to neurons have received increasing attention. The use of iPSCs as an experimental model or as a tool in regenerative medicine requires great attention to the factors and conditions exploited in reprogramming and differentiation protocols [25]. IPSC-derived neurons are widely used models to investigate neurodegeneration; however, our understanding of metabolic changes occurring during the differentiation process towards DA neurons is still poorly detailed. Apart from gene and protein functions that are influenced by epigenetic regulation and post-translational modifications, respectively, metabolites represent direct signatures of biochemical activities, and they are, therefore, a direct readout of the cellular phenotype. On this ground, by the analysis of a large set of metabolites, novel findings have been elucidated that link cellular pathways to biochemical and molecular mechanisms, thus improving our knowledge of physiological and pathological processes occurring in cells, tissues or organisms [26].

In this study, we characterised from a metabolic point of view the differentiation process of iPSCs to DA neurons. One of the key features that we reported is that primed iPSCs are metabolically different from neurons at all the analysed time points (24, 45 and 60 days of differentiation). Specifically, the pluripotent stem cells showed higher fatty acid oxidation and very low levels of lactate production, suggesting that at this stage, they rely more on fatty acid β-oxidation as a fuel source than on glycolysis. Upon progression to neuronal progenitors (24 days of differentiation), it was observed that cells began to shut down fatty acid β-oxidation and preferentially catabolised glucose. Most evident in these neuronal progenitors is the consumption of certain amino acids such as glutamine and alanine, which fuelled, through transamination reactions, several intermediates of the TCA cycle such as α-ketoglutarate, succinate, fumarate and malate. Furthermore, citrate levels peaked at 24 days after differentiation, suggesting that this anabolic molecule may be utilised for the synthesis of lipids, such as cholesterol and fatty acids. In addition, methionine content was also found to be reduced, probably due to its use in the generation of S-adenosyl methionine, an important methyl group donor, crucial for epigenetic regulation, protein functions and other biochemical reactions. Methionine may also be involved in 1-carbon metabolism, useful for the synthesis of nucleotides, such as purines and thymidine, and it is also fundamental in generating other amino acids, such as glycine and cysteine, essential for the biosynthesis of glutathione involved in redox defence [27]. In line with these observations, the levels of ribose-5-phosphate/xylulose-5-phosphate/ribulose-5-phosphate, among which the first is a key molecule involved in nucleotide biosynthesis, were induced in neuronal progenitor cells. This latter observation is of utmost importance considering that the oxidative branch of the pentose–phosphate pathway, besides producing ribose-5-phosphate/xylulose-5-phosphate/ribulose-5-phosphate, also generates two molecules of NADPH. The first NADPH is involved in glutathione regeneration to lower oxidative stress, and the second is utilised in lipid biosynthesis [28]. In this regard, the cytosolic enzyme ATP-citrate lyase breaks down citrate into acetyl-CoA and oxaloacetate. Acetyl-CoA is then available for cholesterol and fatty acid synthesis, while oxaloacetate is converted into malate by cytosolic malate dehydrogenase 1. Next, malate, by the action of the malic enzyme, is transformed into pyruvate that re-enters the TCA cycle, and the NADPH released is utilised to sustain lipid synthesis together with that coming from the pentose–phosphate pathway.

As demonstrated by principal component analysis, differentiated cells at 45 days shared common features with those differentiated for 60 days. At both time points, cells were highly glycolytic, as demonstrated by increased levels detected of DHAP/GAP, PEP and above all, lactate. Furthermore, TCA cycle intermediates reached a steady state, and several amino acids were found to have increased, including those needed for dopamine synthesis such as phenylalanine and tyrosine.

In summary, our study provides, for the first time, a metabolic characterisation during the differentiation process of iPSCs to DA neurons, highlighting the metabolic rewiring occurring during the different stages of neuronal differentiation. Remarkably, it is possible that different cell types or even different neurons may need to activate or inhibit different metabolic pathways than those described here to achieve full differentiation. This is of utmost importance considering that shifts in metabolism drive changes in epigenetic configuration that ultimately affect chromatin dynamics and determine gene expression in a given cell at some stage during the differentiation process.

## Figures and Tables

**Figure 1 biomedicines-10-02069-f001:**
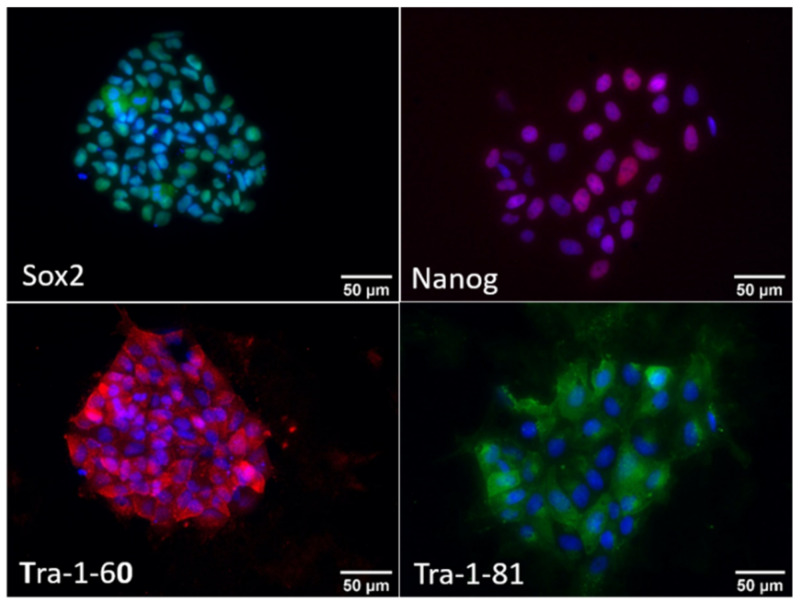
Characterisation of iPSCs. Representative immunofluorescence images of human iPSCs stained for the pluripotent markers Sox2 (green, (**upper panel**)), Nanog (red, (**upper panel**)), Tra-1-60 (red, (**lower panel**)) and Tra-1-81 (green, (**lower panel**)). Cell nuclei were stained with Hoechst (blue). Images were acquired at 400× magnification.

**Figure 2 biomedicines-10-02069-f002:**
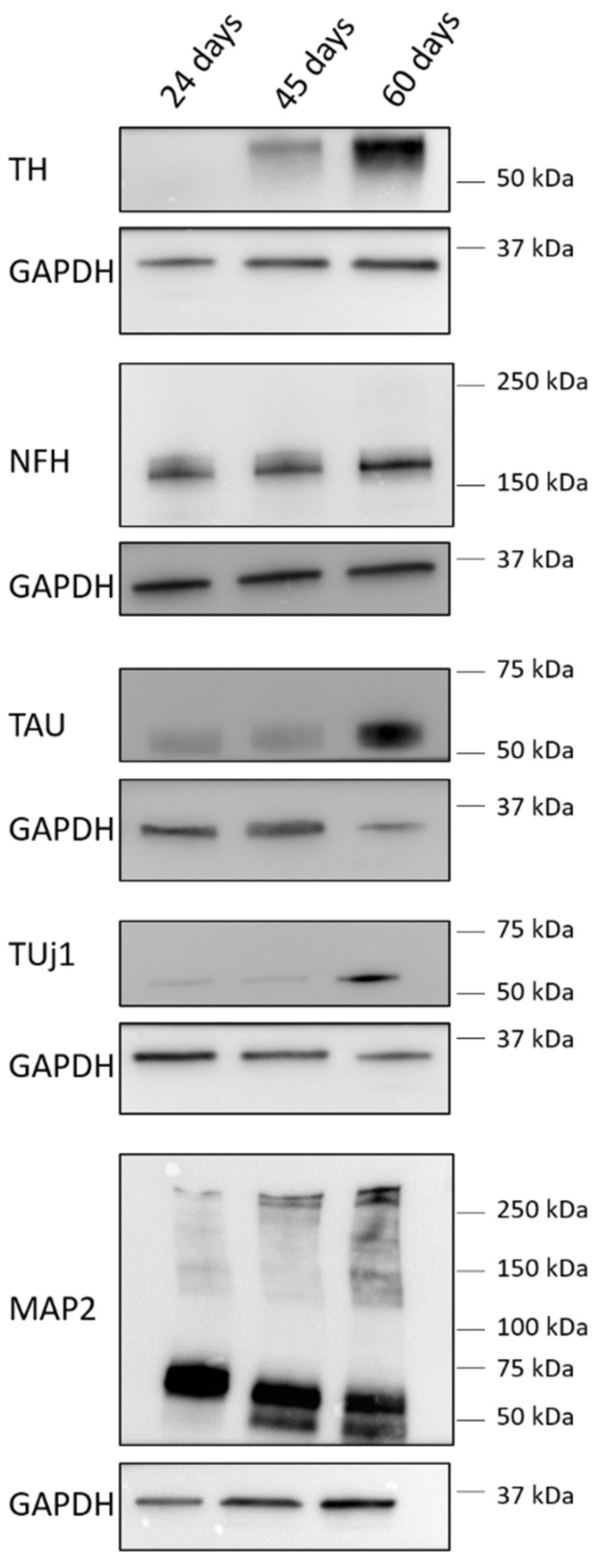
Evaluation of neuronal markers’ expression during differentiation of iPSCs into dopaminergic neurons. Representative immunoblotting images of the expression of tyrosine hydroxylase (TH), neurofilament H (NF-H), TAU, β-III-tubulin (TUJ1) and microtubule-associated protein 2 (MAP2) in iPSCs differentiated into DA neurons for 24, 45 and 60 days. Housekeeping GAPDH expression has been used as the loading control.

**Figure 3 biomedicines-10-02069-f003:**
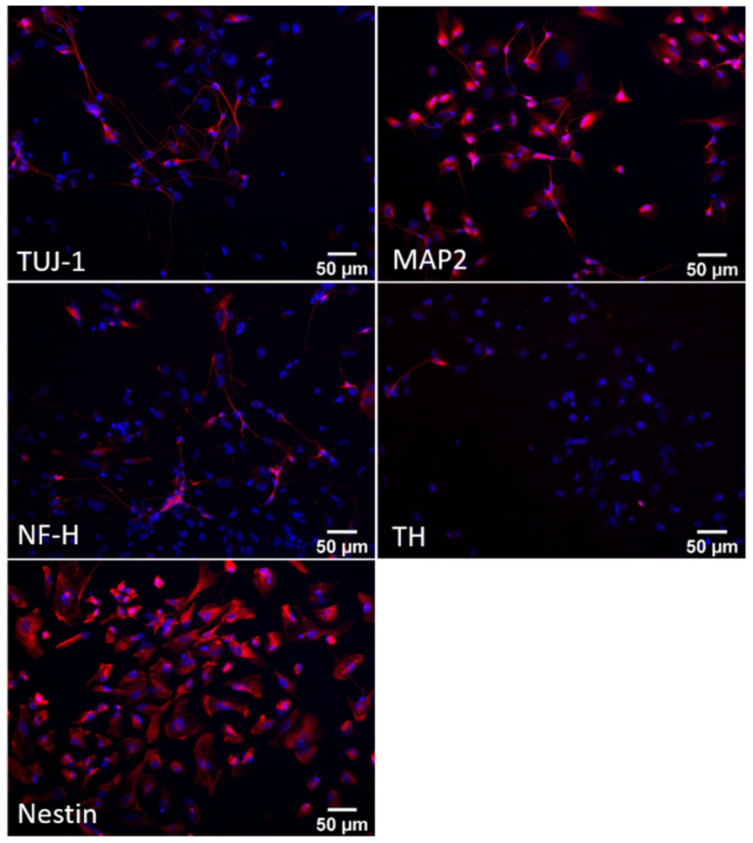
Characterisation of iPSCs differentiated for 24 days into dopaminergic neurons. Representative immunofluorescence images of human iPSC-derived dopaminergic neurons at day 24 of differentiation. Cells were stained (red) for the neuronal markers β-III-tubulin (TUJ1), neurofilament H (NF-H), microtubule-associated protein 2 (MAP2) and tyrosine hydroxylase (TH) and for the neuronal precursor marker nestin. Cell nuclei were stained with Hoechst (blue). Images were acquired at 200× magnification.

**Figure 4 biomedicines-10-02069-f004:**
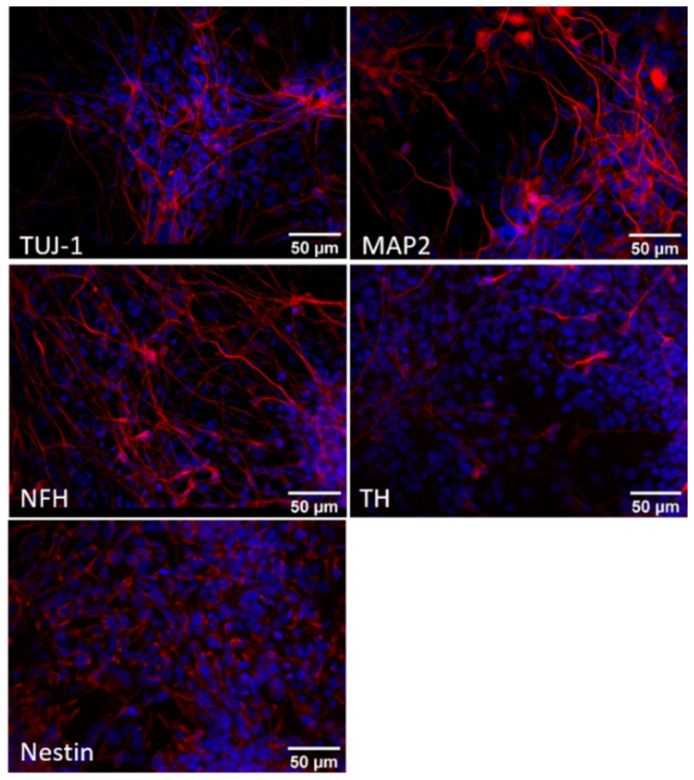
Characterisation of iPSCs differentiated for 45 days into dopaminergic neurons. Representative immunofluorescence images of human iPSCs-derived dopaminergic neurons at day 45 of differentiation. Cells were stained (red) for the neuronal markers β-III-tubulin (TUJ1), neurofilament H (NF-H), microtubule-associated protein 2 (MAP2) and tyrosine hydroxylase (TH) and for the neuronal precursor marker nestin. Cell nuclei were stained with Hoechst (blue). Images were acquired at 400× magnification.

**Figure 5 biomedicines-10-02069-f005:**
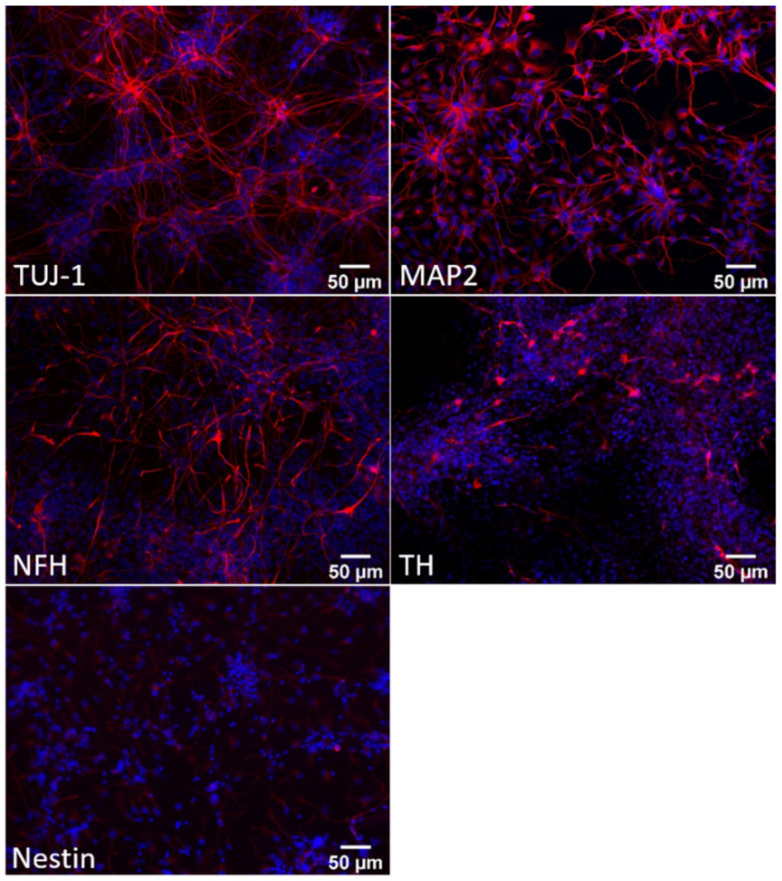
Characterisation of iPSCs differentiated for 60 days into dopaminergic neurons. Representative immunofluorescence images of human iPSC-derived dopaminergic neurons at day 60 of differentiation. Cells were stained (red) for the neuronal markers β-III-tubulin (TUJ1), neurofilament H (NF-H), microtubule-associated protein 2 (MAP2) and tyrosine hydroxylase (TH), and for the neuronal precursor marker nestin. Cell nuclei were stained with Hoechst (blue). Images were acquired at 200× magnification.

**Figure 6 biomedicines-10-02069-f006:**
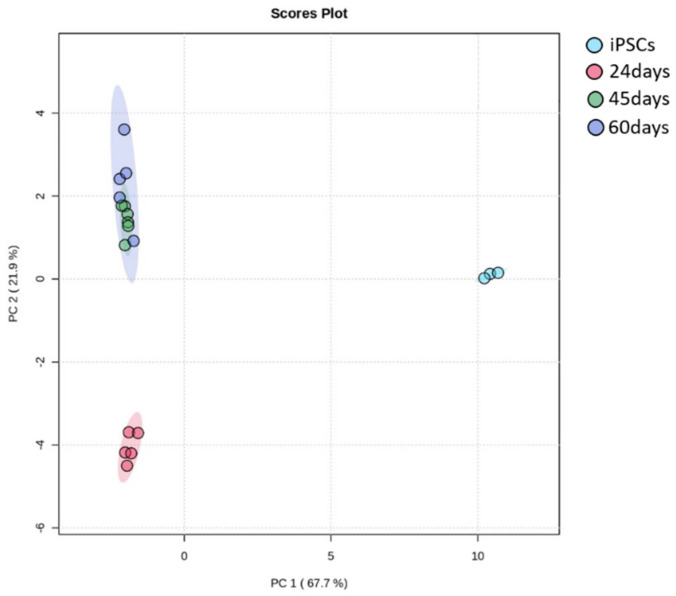
Principal component analysis (PCA) from metabolomics of iPSCs and dopaminergic neurons at days 24, 45 and 60 of differentiation.

**Figure 7 biomedicines-10-02069-f007:**
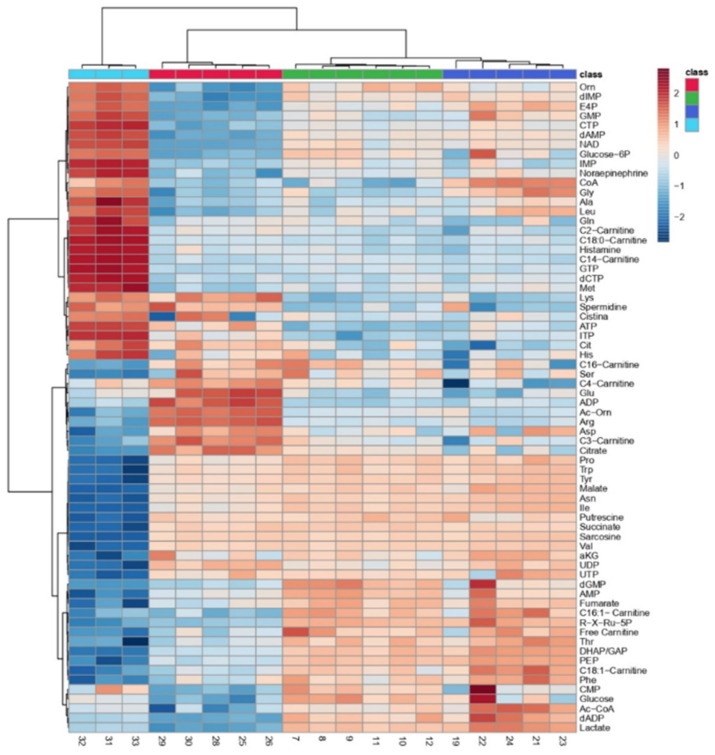
Heatmap of metabolites detected by mass spectrometry in iPSCs and dopaminergic neurons at days 24, 45 and 60 of differentiation.

**Figure 8 biomedicines-10-02069-f008:**
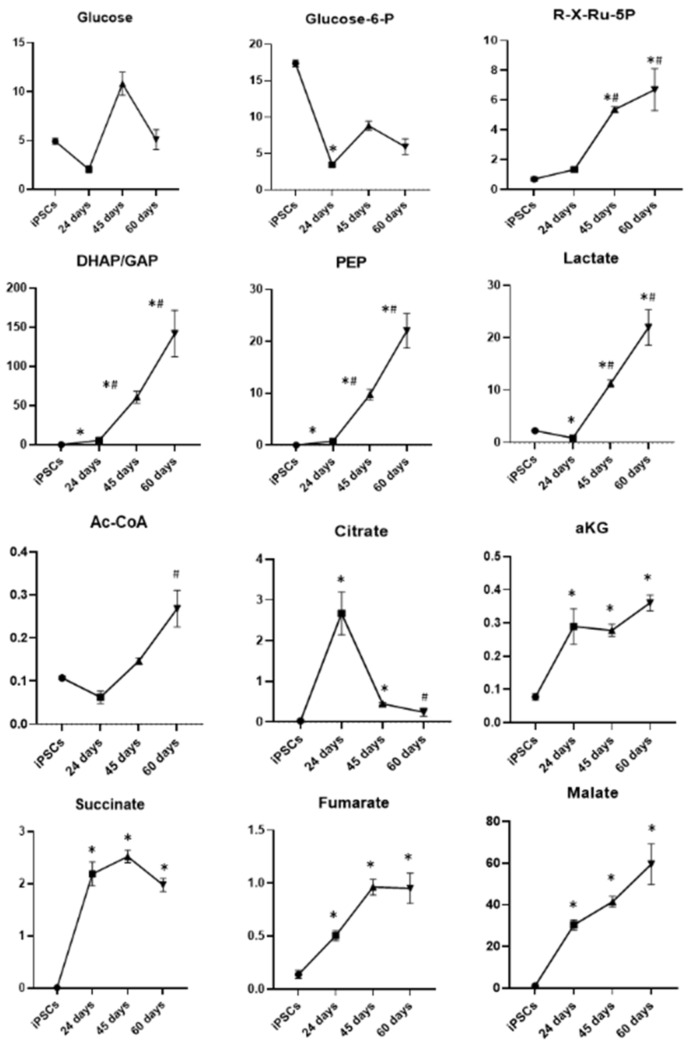
Graphic representation of the major metabolic changes related to glycolysis, pentose–phosphate pathway and tricarboxylic acid cycle in iPSCs and DA neurons at days 24, 45 and 60 of differentiation: glucose-6-phosphate, glucose-6-P; DHAP/GAP, dihydroxyacetone phosphate/glyceraldehyde 3-phosphate; PEP, phosphoenolpyruvate; R-X-Ru-5P, ribose-5-phosphate/xylulose-5-phosphate/ribulose-5-phosphate; acetyl-CoA, acetyl coenzyme A; αKG, α-ketoglutarate. Data represent mean ± SEM from at least triplicates of cultures from iPSCs and DA neurons at days 24, 45 and 60 of differentiation. * Significant v. iPSCs, ^#^ significant v. 24 days.

**Figure 9 biomedicines-10-02069-f009:**
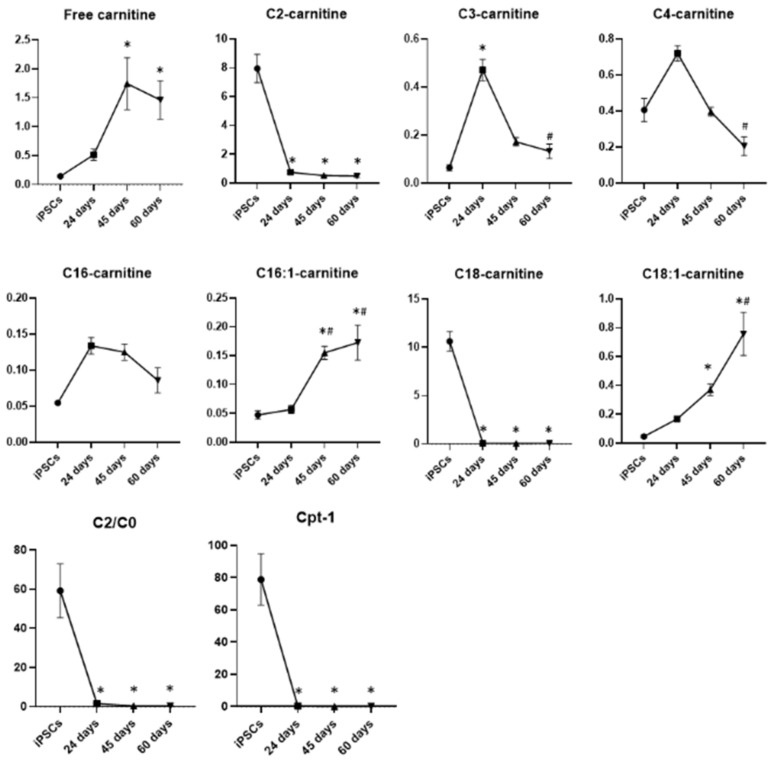
Graphic representation of the major metabolic changes related to fatty acid β-oxidation in iPSCs and DA neurons at days 24, 45 and 60 of differentiation. Data represent mean ± SEM from at least triplicates of cultures from iPSCs and DA neurons at days 24, 45 and 60 of differentiation: C0, free carnitine; C18-carnitine, ocatadecanoyl carnitines; C16-carnitine, hexadecanoylcarnitines; C2-carnitine, acetylcarnitines; C3-carnitine, propionylcarnitines; C4-carnitine, butyrylcarnitines; C18:1-carnitine, octadecenoylcarnitine; C16:1-carnitine, hexadecenoylcarnitine; Cpt-1, carnitine palmitoyl transferase 1. * Significant v. iPSCs, ^#^ significant v. 24 days.

**Figure 10 biomedicines-10-02069-f010:**
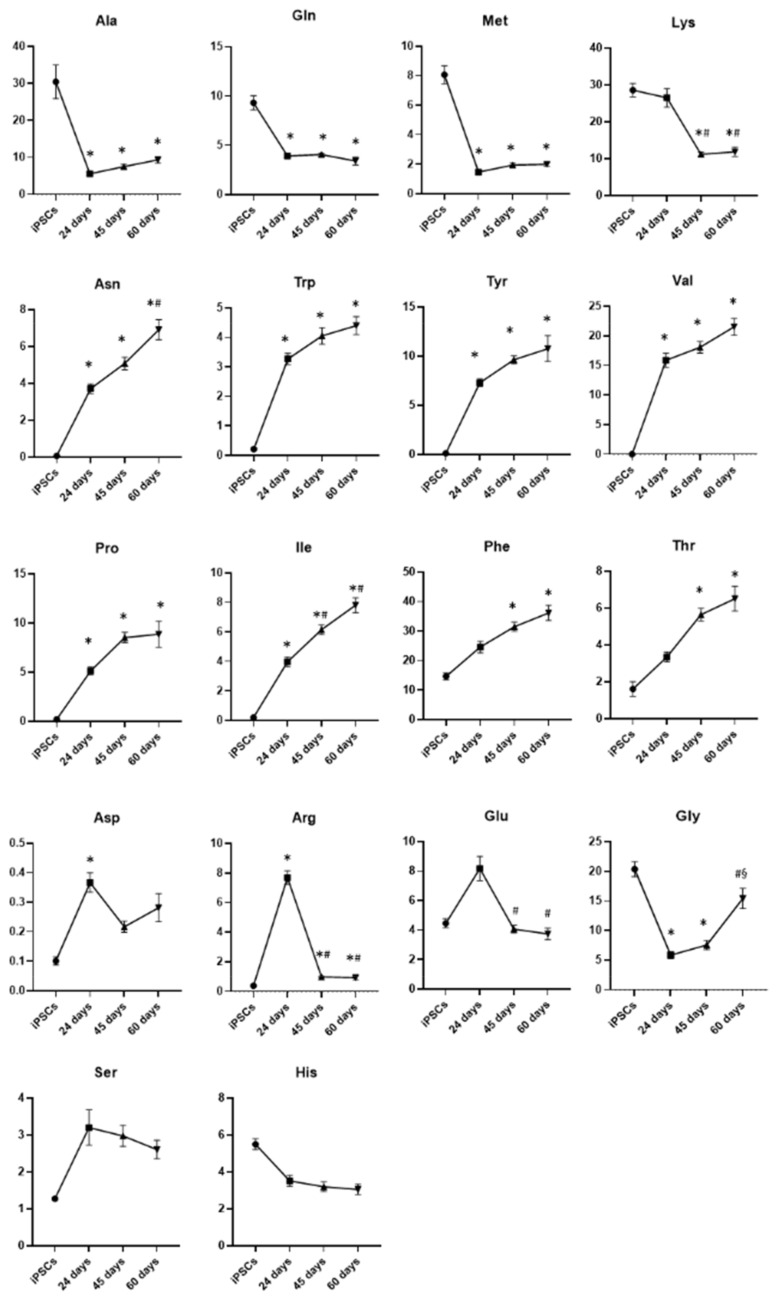
Graphic representation of the major changes related to amino acids in iPSCs and DA neurons at days 24, 45 and 60 of differentiation. Data represent mean ± SEM from at least triplicates of cultures from iPSCs and DA neurons at days 24, 45 and 60 of differentiation: Ala, alanine; gln, glutamine; met, methionine; lys, lysine; asn, asparagine; trp, tryptophan; tyr, tyrosine; val, valine; pro, proline; ile, isoleucine; phe, pheylalanine; thr, threonine; asp, aspartate; arg, arginine; glu, glutamate; gly, glycine; ser, serine; his, histidine. * Significant v. iPSCs, ^#^ significant v. 24 days, ^§^ significant v. 45 days.

## Data Availability

The data sets generated and analysed during the current study are available from the corresponding author on reasonable request.

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
