# Peer review of "Metabolic Profile Variations along the Differentiation of Human-Induced Pluripotent Stem Cells to Dopaminergic Neurons"

_biomedicines, 2022, doi:10.3390/biomedicines10092069_

Round 1
Reviewer 1 Report
This paper aims to provide s for the first time a metabolic characterization during the differentiation process of iPSCs to DA neurons, highlighting the metabolic rewiring occurring during the different stages of neuronal differentiation.
The manuscript is written comprehensively enough to be understandable despite of the complexity of the subject.
They addressed this aim by performing metabolomics analysis of cells at different stages of differentiation. They found that iPSCs are metabolically different from neurons at all the analysed time points.
The paper stated the purpose, discussion and global implication are clearly stated and consistent with the rest of the manuscript; authors provided enough information in their discussion by using a good number of important articles talked about the subject.
The authors clearly elucidated that it is possible that different cell types or even different neurons may need to activate or inhibit different metabolic pathways than those described in the paper to achieve full differentiation. The authors addressed their hypothesis and opinion in a reproducible way and proved their results through all the required experiments and analysis and they used enough number of analyses to prove their results. The results were presented in a clear way which facilitate in reaching a conclusion.
The abbreviations should be explained at the first place they are mentioned.
In vitro, in vivo: should be written in italic
No plagiarism has been detected.
References: The authors should follow the journal guidelines.
Author Response
We thank the reviewer for the kind comment and for appreciating our work. We addressed the comments as follows:
Comment: The abbreviations should be explained at the first place they are mentioned.
Answer: We checked the manuscript and figure legends for abbreviations and mentioned them at the first place they appeared. All the changes are tracked.
Comment: In vitro, in vivo: should be written in italic
Answer: We did it as suggested. All the changes are tracked.
Comment: References: The authors should follow the journal guidelines.
Answer: We checked the references to be in accordance with the journal guidelines.
Reviewer 2 Report
This is a very interesting study concern the alteration of metabolic profile in iPSC differentiated from precursor of neuron to dopaminergic neurons. The determination method was based on liquid chromatography-tandem mass spectrometry. The results
showed that iPSCs primarily rely on fatty acids beta-oxidation and shift to glucose catabolization when differentiated to dopaminergic neurons. Furthermore, the content of amino acid metabolization was also influenced by the differentiation.
But the quality of immunohistochemistry staining in Figure 1, 3, 4, and 5 were poor, it is better to take photography using confocal imaging which gain the better resolution.
Author Response
We thank the reviewer for the kind comment.
Comment: The quality of immunohistochemistry staining in Figure 1, 3, 4, and 5 were poor, it is better to take photography using confocal imaging which gain the better resolution.
Answer: We agree with the reviewer comment that confocal imaging would allow a better resolution, however here we want to give a general idea about the differentiation status of the neurons, which we should be represented by the immunofluorescence images in Figure 1, 3, 4, and 5. To further validate the changes in expression of neuronal and dopaminergic markers, we included also immunoblotting analysis.
Reviewer 3 Report
The originality of the topic is excellent. The work is a significant contribution to the field, and it's well organized. All the parts are properly and comprehensively described. The possible clinical implications (Parkinson's disease) appear very clear.
Author Response
We thank the reviewer for the kind comment and for appreciating our work.